# The Effect of Mineral Admixtures and Fine Aggregates on the Characteristics of High-Strength Fiber-Reinforced Concrete

**DOI:** 10.3390/ma15248851

**Published:** 2022-12-12

**Authors:** Olga Vladimirovna Aleksandrova, Nguyen Duc Vinh Quang, Boris Igorevich Bulgakov, Sergey Viktorovich Fedosov, Nadezhda Alekseevna Lukyanova, Victoria Borisovna Petropavlovskaya

**Affiliations:** 1Institute of Industrial and Civil Engineering, National Research Moscow State Civil Engineering University, Yaroslavskoe Shosse, 26, 129337 Moscow, Russia; 2Hue Industrial College, 70 Nguyen Hue Street, Hue City 530000, Vietnam; 3Faculty of Civil Engineering, Tver State Technical University, Af. Nikitin, 22, 170026 Tver, Russia

**Keywords:** high-strength concrete, active mineral additives, fine aggregate, particle packing density, finely ground natural quartz sand

## Abstract

Introduction: the article discusses the effect of the complex of active mineral additives consisting of silica and fly ash, and a fine aggregate, including finely ground natural-white quartz sand for partial replacement of river sand, on the mechanical properties of high-strength concrete containing steel fiber. Materials and methods: high-strength concrete containing Dramix^®^3D 65/35 steel fiber in the amount of 100 kg per 1 m^3^ of concrete mixture was suggested where 22% to 100% of river sand was replaced by finely ground white natural sand of the particle size of 5 to 1800 μm and containing the complex of active mineral additives for partial replacement of cement as part of a multicomponent binder, consisting of low-calcium fly ash of thermal power plants and silica and containing, respectively, 20, 30, 40% fly ash and from 5 to 15% silica by weight of the binder. Results: research results have shown that 100% replacement of river sand with finely ground natural white sand, in concrete containing 20% of the mass as part of a multicomponent binder, fly ash and from 5 to 15% by weight of silica, contributes to the increase of its strength properties: the values of concrete compressive strength after 28 days were in the range from 118.5 to 128 MPa, tensile strength during bending and splitting, respectively, from 18.8 to 25.4 MPa and from 10.2 to 11.9 MPa, which is higher than the strength of concrete samples containing river sand. Conclusions: the achieved results have demonstrated the efficiency of using finely ground natural white sand as an alternative to river sand for producing high-strength concrete, thus helping to save the river sand resources in Vietnam. The use of fly ash and micro silicon, which are power and metallurgy wastes, as part of a multicomponent binder in order to partially replace cement reduces the carbon footprint in the production of binders and will also have a beneficial effect on environmental protection against industrial waste pollution.

## 1. Introduction

High-strength concrete is characterized by high compressive strength and is comparable to such high-strength natural rocks as basalt [1,2,3]. The use of high-strength concrete in construction can significantly reduce the cross-section of structures, make them lightweight and delicate, as well as reduce the costs of raw materials and of transporting finished products. The unique combination of the properties of such concretes makes it possible to obtain thin profiles and shells, and complex and various geometric shapes of products and structures, without losing their strength and durability. Over the past two decades, there has been a growing interest in high-strength concretes, while their scope is expanding, from repair and restoration of reinforced concrete structures and architectural elements to the oil and gas industry applications as well as to the construction of various hydraulic and underground structures [4]. Today, the most popular use of high-strength concrete is for the construction of bridges [5,6,7,8].

Important characteristics of high-strength concretes are strength, durability and economy. High strength can be achieved by adding finely dispersed materials with pozzolanic properties to the concrete mixture (for example, micro silica and fly ash from thermal power plants), modern super- and hyper plasticizers, by the use of micro-reinforcing fibers (metal or synthetic), low water-to-binder ratio, increased cement content, as well by the longer mixing during concrete mix preparation and heat treatment of products and structures.

High-strength fiber-reinforced concrete (HSFRC) can be viewed as the result of combining three concrete technologies: self-compacting concrete (SCC), fiber-reinforced concrete (FRC), and high-performance concrete (HPC) [9,10]. High-strength fiber-reinforced concrete is a high-density steel fiber reinforced composite material with a compressive strength of over 170 MPa, an axial tensile strength of over 8 MPa, and a flexural tensile strength of over 30 MPa [11,12].

The high strength of high-strength concrete is achieved by improving its production technology and the used materials (by adding ultra-thin pozzolans), as well as by a very low water–cement ratio, the use of highly effective superplasticizers, a high binder content, and an optimal volume of high-strength steel fibers. The use of steel fiber is aimed at preventing the growth and development of microcracks by reducing tensile stresses, which prevents their association into macrocracks [13,14]. The example of a concrete mix composition used for obtaining high-strength concrete is presented in [15,16]. High binder content is necessary to obtain the minimum compressive strength of 150 MPa for high-strength concrete.

In the concrete mixture, the binder content is almost 40% of its total. At the same time, micro silica accounts for almost 25% of the binder mass, and its amount by mass can reach 30% [17,18]. The use of micro silica is necessary to achieve high compressive strength and durability of concrete. Micro silica enters a pozzolanic reaction with free calcium hydroxide (portlandite), resulting in the formation of additional low-basic calcium hydrate silicates (C-S-H), filling the voids in the cement stone and, thereby, compacting the concrete structure [19]. However, the improvement in the concrete properties caused by adding micro silica comes at a price: in the modern commodity market, micro silica cost is 4–7 times higher than Portland cement. Wang et al. [20] stated that high-strength concrete with the minimum compressive strength of 138 MPa after 28 days and 150 MPa after 56 days can be obtained by replacing 10% Portland cement with micro silica in the binder composition. Similarly, El-Hadj Kadri et al. [21] concluded that the effect of micro silica on the compressive strength of concrete gives the optimal result when replacing 10% of cement, and in contrast, when its content is increased to 20 and 30%, a decrease of concrete strength is observed. At the same time, the effect of micro silica and other pozzolanic materials on the strength properties of concrete may depend on their curing conditions. In this study, the authors define the most efficient micro silica content in the binder composition able to produce high-strength concrete using locally available materials, thus reducing the concrete cost.

Quartz flour is another fine filler, which accounts for 8.4% of the total mass of the mixture. Quartz flour has an average particle diameter slightly smaller than that of Portland cement, which allows it to fill possible voids between sand grains, cement particles, and hydration products and thereby seal the structure of concrete cement stone, which increases its strength and reduces permeability. However, the use of finely ground quartz materials may not be required due to a significant proportion of non-hydrated Portland cement, which fills voids and forms a dense matrix of cement stone. Karine V and Maximilien et al. [22] found that the rigidity of non-hydrated cement particles is greater than that of other finely dispersed components of the concrete mixture. Consequently, the water-to-binder ratio (W/B) can be reduced as long as there is a sufficient amount of hydration products to bind all the components of the concrete mixture into a solid matrix. This makes it possible to exclude quartz powder from the composition of the mixture to further reduce the cost of concrete.

For high-strength concretes, the filler is an important component, as its content is almost 50% from the mass, and affects the concrete’s strength and cost. The grain size of the aggregates also has a general effect on the concrete’s compressive strength. The composition presented in Table 1 uses fine sand (150–600 μm) to ensure the uniformity of the concrete and to increase the strength.

**Table 1 materials-15-08851-t001:** Chemical composition of binding materials.

Materials	Chemical Composition, %							
	SiO_2_	Al_2_O_3_	Fe_2_O_3_	CaO	MgO	K_2_O	Na_2_O	TiO_2_
Finely ground quartz powder (Qp)	99.77	0.045	0.039	0.051	0.037	0.006	0.02	0.04
Silica fume (SF)	92.50	0.87	1.92	0.31	0.84	1.23	0.39	-
Fly ash Pha Lai (FA)	57.53	24.06	6.06	0.69	0.97	3.56	0.28	0.67
Cement PCsr40 (C)	21.45	3.76	5.07	63.22	1.71	0.63	0.13	0.12
Continuation of Table 2								
Finely ground quartz powder (Qp)	P_2_O_5_	ZrO_2_	Cr_2_O_3_	CuO	ZnO	BaO	SO_3_	L.O.I. *
Silica fume (SF)	-	-	-	-	-	-	-	0.042
Fly ash Pha Lai (FA)	0.12	0.029	0.019	0.021	0.029	-	0.3	5.64
Cement PCsr40 (C)	-	-	-	-	-	1.88	2.0	-

* L.O.I.—Loss on ignition.

**Table 2 materials-15-08851-t002:** Physical properties of steel fiber Dramix^®^3D.

Indicators	Diameter, D, mm	Length, L, mm	Ratio L/D	Tensile Strength, MPa	Elastic Modulus, MPa	Density, kg/m^3^	Appearance
Value	0.55	35	≈65	1300	200,000	7850	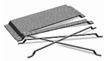

The use of water-reducing superplasticizer (HRWR) is necessary to obtain high-strength concrete and its dosage can vary Gerlicher et al. [23] determined the dosage of 35 kg/m^3^ of superplasticizer to be suitable for producing concrete mix with 360 mm cone flow.

It is of interest to study the possibility of obtaining concrete containing steel fiber and having high strength by using a developed complex of active mineral additives consisting of silica and fly ash, as well as a fine aggregate including finely ground natural white quartz sand, still little used, in order to partially replace the scarce river sand in Vietnam, and to investigate the mechanical properties of the resulting high-strength concrete. In addition, the use of fly ash and silica, which are energy and metallurgy wastes, as part of a multicomponent binder for the purpose of partial replacement of cement will help reduce the carbon footprint in the production of binders and will also have a beneficial effect on environmental protection from industrial waste pollution.

## 2. Materials and Methods

### 2.1. Pilot Study

High-performance concretes have a high particle packing density contributing to low porosity, high mechanical strength, and impermeability. Figure 1 shows the compositions of concretes with different strength values.

The composition of high-performance concretes, in addition to cement, fine aggregates, and additives, also includes steel fiber and finely dispersed nanofillers, which, in terms of achieving high strength to compressive and tensile loads, replace coarse aggregates in their composition [24].

The most widely used theoretical model for creating the maximum dense packing of particles in the design of UHPC is the model by Anderson and Andreasen [12]. However, it considers particles only under dry conditions, which may not reflect the actual packing of UHPC particles, since the effect of water and other liquids is not taken into account [25]. Therefore, the wet particle packing density method was introduced to obtain the “real” maximum packing of particles [26]. In this paper, UHPC design methods are considered based on dry particle packing and wet particle packing models.

### 2.2. Materials

The following raw materials were used in the work.

The binder is the sulfate-resistant Portland cement PCsr40 manufactured by Luks Cement Ltd. (Hue city, Vietnam) with 3 and 28-day compressive strength, respectively, 34 MPa and 50 MPa, the setting onset of 130 min and the setting end of 170 min, 2.49% C_3_A content and 21.23% C_4_AF + C_3_A, the density of 3150 kg/m^3^, meeting the requirements of TCVN 7711 [27] and GOST 22266 [28] standards.

Active mineral additives with pozzolanic reactivity and facilitating structure densification are: micro silica Sikacrete^®^PP1–Sika Limited (Nhơn Trạch, Vietnam), with the particle size of <0.1 µm, the true density of 2150 kg/m^3^ according to TCVN 8827-2011 [29] and GOST R 56592 [30], as well as the class F fly ash from Pha Lai Thermal Power Plant (Chi Linh, Vietnam), according to TCVN 10302-2014 [31] and GOST 25818 [32]. The chemical composition of cement and mineral additives is presented in Table 1.

Fine aggregate-quartz sand from the Huong River, with the true density of 2650 kg/m^3^ and the particle size modulus of 2.91; natural white sand with the average particle size of 0.1–1.8 mm compliant with the requirements of TCVN 7570-2006 [33], TCVN 10796-2015 [34], and GOST 8736-2014 [35], and finely ground quartz powder (quartz flour) obtained by grinding white sand with the average particle size of 5–63 μm, which is a microfiller. Particle size distribution curves for these materials are shown in Figure 2.

Superplasticizer (SP) Sika^®^ViscoCrete^®^151–Sika (Vietnam), with the density of 1.09 kg/L, according to TCVN 10302-2014 [36] and GOST 25818 [37].

Dramix^®^3D 65/35 steel fiber with 0.55 mm diameter (D), 35 mm length (L), and L:D ratio ≈ 65 (Table 2) according to TCVN 12392-1: 2018 [38].

Drinking water for preparation of concrete mixtures.

### 2.3. Research Methods

1.Packing method for dry particles

Fuller [39] and Andersen [40] developed the first continuous model by introducing the target particle size distribution P(D). Taking into account the effect of the minimum particle size on their packing, Funk and Dinger [41] developed a modified model and introduced *D_min_*. To obtain high-strength concrete, it is necessary for its frame to be dense and durable. When designing high-strength concrete, a continuous model is more preferable, making possible to obtain a denser framework of particles [42].

In this study, this modified model was used to design the specific matrix in accordance with Equation (1):(1)$$P\left(D_{i}\right)=\left(D_{i}^{q}-D_{min}^{q}\right)/\left(D_{max}^{q}-D_{min}^{q}\right)$$
where: *P*(*D_i_*)—the percentage of particles that can pass the sieve with smaller than *D*; *D_i_*—average particle size, mm; *D_max_*—maximum particle size, mm; *D_min_*—minimum particle size, mm; *q*—the particle size distribution modulus.

Using Equation (1), it is possible to design different concrete compositions, taking different values of q, which determines the percentage ratio between small and large particles in the concrete mixture. Brouwers and Radix [43] suggested that the distribution modulus is within the range of 0–0.25 for concrete with a high content of binders [44]. For high-strength concrete, the value of q was proposed within the range of 0.21–0.25 [45]. Using the modified model, Yu et al. developed the concrete mixture compositions for obtaining high-performance concretes [46]. At the same time, the distribution modulus was set at 0.23. Given the fact that large numbers of small particles are used to obtain the matrix of high-strength concrete, based on the recommendation [44], the value of q in this study was fixed at 0.23. The mass fractions of each individual raw component in the concrete mixture are adjusted to achieve the optimal ratio (the smallest difference) between the formulated mixture (Figure 3, dotted thick line) and the target curve (Figure 3, solid thick line), using the algorithm based on the “Method of least squares”, as shown in Equation (2) [47]:(2)$$RSS=\;\frac{\sum_{i=1}^{n}{\left(P_{mix}\left(D_{i}^{i+1}\right)-P_{target}\left(D_{i}^{i+1}\right)\right)}^{2}}{n}\;\Rightarrow Minimum$$
where: *RSS*—residual sum of squares (at defined particle sizes); *P_mix_*—composed mixture; *P_target_*—the target mix of the granulometric composition calculated by Equation (1); n is the number of points (between *D_min_* and *D_max_*) used to calculate the deviation.

The quality of the fit of the resulting cumulative particle size distribution curve is evaluated by using the determination factor *R*^2^, as shown in Equation (3), since this factor provides the correlation value between the gradation of the target curve and the represented concrete mix:(3)$$R^{2}=1-\frac{\sum_{i=1}^{n}{\left(P_{mix}(D_{i}^{i+1})-P_{target}\left(D_{i}^{i+1}\right)\right)}^{2}}{\sum_{i=1}^{n}(P_{mix}{\left(D_{i}^{i+1}-{\bar{P}}_{mix}\right)}^{2}}$$
where ${\bar{P}}_{mix}\;=\;\frac{1}{n}\sum_{i=1}^{n}P_{mix}(D_{i}^{i+1})$—the average value of the entire distribution.

However, the dry particle packing method does not account for the effects of water and HRWR. In practice, water and superplasticizer play an important role in particle packing and therefore affect the properties of UHPC [48].

2.Packing method for wet particles

The high friction of dry particles prevents their packing density increase [49]. The presence of water reduces the friction force, and if the particles are in a water-saturated or supersaturated state, then this friction force can be eliminated [50]. Moreover, adding superplasticizer affects the thickness of the water film formed on the surface of solid particles [48].

Therefore, when using the model of wet packing of particles, higher packing density will be achieved as compared to the dry packing model (Figure 4), since taking the moisture content of particles into account allows to develop the model that is more accurate and closer to reality [51].

The high wet packing density improves the macro002Dmeso-micropore structure, resulting in the increase of the compressive strength of UHPC [48]. The packing model of wet particles was proposed accounting for the effect of water and superplasticizer. The following is necessary for achieving the dense packing of wet particles: (1) setting the initial value of W/B; (2) measuring the required amount of water and cementing materials and mixing them; (3) placing the resulting mixture into cylindrical mold and determining the mass of the cement paste; (4) calculating the solids concentration (φ) and void ratio (u) using Equations (4)–(6); (5) repeating the above steps at the lower W/B ratio until the maximum packing density is reached.
(4)$$V_{c}=\frac{M}{\rho_{w}u_{w}+\rho_{\alpha }R_{\alpha }+\rho_{\beta }R_{\beta }+\rho_{\gamma }R_{\gamma }}$$
(5)$$u\;=\left(V-V_{c}\right)/V_{c}$$
(6)$$\varphi \;=V_{c}/V$$
where: M and V are the mass and volume of cement paste in cylindrical form (the form has 62 mm diameter and 60 mm height); ρ_w_ is the water density, ρ_α_, ρ_β_, and ρ_γ_ are the densities of the solid components of various binders; R_α_, R_β_, and R_γ_—volumetric ratios of the solid components of various binders.

The authors of [25] shows an example of determining the minimum void ratio (u) and the maximum solids concentration (ϕ). The optimum W/B can be determined by considering the maximum wet pack density. However, the highest packing density of particles does not always result in the expected properties of UHPC. For example, high particle packing density does not provide high fire resistance of concrete, since relatively high porosity is preferable for reducing pore pressure in high temperature environment.

The developed compositions of high-strength concretes using industrial waste are presented in Table 3.

### 2.4. Preparation of Steel Fiber Concrete Mixture and Molding of Samples

In order to obtain UHPC, it was necessary to follow the following sequence of steps in preparing the concrete mixture: (1) fine aggregates were mixed in a dry state for 2 min for preventing agglomeration, as well as evenly distributing fine particles; (2) thereafter, adding binders, including cement, fly ash, and micro silica, followed by 5 min of mixing; (3) steel fibers were then gradually added and stirred for another 3 min until these fibers have been completely dispersed; (4) then, continuing stirring for about 10 min, gradually adding the solution of superplasticizer previously mixed with water to the resulting mixture; (5) the workability of the prepared concrete mixture was determined by the flow of the cone, which should be in the range from 280 to 310 mm without delamination, and then filled with the concrete mixture of the form.

The consistency of the concrete mixture was determined according to TCVN 12209-2018 standards [52] and GOST R 57812-2017/EN 12350-5:2009 [53]. The molds with the samples covered with a damp cloth were stored for 24 h in the open air at the temperature of 20 ± 5 °C, then the samples were removed from the molds and placed in the tank with fresh water of 20 ± 2 °C temperature where they hardened until reaching the estimated age of 3, 7, 14, 21, 28, and 120 days. The samples prepared in this way had the form of cubes 100 × 100 × 100 mm size for compressive strength tests, cylinders 100 × 200 mm for axial tensile tests and prisms 100 × 100 × 400 mm for tensile tests in bending.

## 3. Results and Discussion

### 3.1. Determination of the Strength Characteristics of the Developed Concretes

The test results are illustrated in Figure 5, Figure 6 and Figure 7. The results of the experiment show the efficiency of combining active mineral additives (micro silica and fly ash) with natural white sand, finely ground quartz powder, and steel fiber in a concrete mixture, allowing to obtain concretes with high strength characteristics, such as compressive strength at 3, 7, 14, 21, 28, and 120 days of hardening, as well as the axial tensile strength and flexural tensile strength at the age of 28 days.

### 3.2. Compressive Strength of Different Curing Ages

We consider the change in compressive strength of the studied concretes with and without natural white quartz sand, obtained as the result of hardening for the concrete mixtures of twelve developed compositions. These compositions containing steel fiber (Table 3), depending on the content of fly ash and micro silica in the composition of a multicomponent binder as a partial replacement of sulfate-resistant Portland cement and hardening tested at different ages are shown on Figure 5 and Figure 6.

Using fly ash has reduced the compressive strength values as compared to the test composition (UC) at the early age of concrete hardening, which can be explained by the cement content decrease in the binder composition. It can be seen that the increase in the fly ash content is accompanied by the increase in the compressive strength of concrete samples at the longer age, namely, the compressive strength decrease at the age of 7 days ranged from 13 to 39%, while, in contrast, at the hardening age of 120 days, we observe the strength increasing by 5–18% as compared to the test concrete composition containing multicomponent binder from 20 to 40% from the mass fly ash to replace part of the sulfate-resistant Portland cement. This indicates the gradual increase in the strength of such concretes in longer hardening periods. This trend can be probably explained by the cumulative effect of cement hydration in combination with the pozzolanic reaction of the used fly ash, leading to a gradual increase in the content of low-basic calcium hydro silicates in the composition of the concrete cement stone [54,55,56]. From the diagram in Figure 5, it can be seen that using natural white sand instead of river sand made a significant impact on the development of early compressive strength of the designed concretes. The results showed that for concretes containing river sand as a fine aggregate, the compressive strength at the age of 3 days was from 64 to 73.5 MPa, and for concretes with white sand, the three-day compressive strength reached 95–110 MPa, meaning the compressive strength of concrete increasing from 29 to 73% at 3-day hardening age. The strength increase at longer age (120 days) was significantly less and ranged from 5 to 29%.

Figure 5 also shows the relationship between the micro silica content in the multicomponent binder composition and the compressive strength increase in the developed concretes with white and river sand at different hardening ages. At the same time, the concrete strength increase was observed with the increase of micro silica content, which can be explained by the increase in the formation of low-basic calcium hydro silicates (CSH) due to the pozzolanic reaction between free calcium hydroxide (CH) and amorphous micro silica (SiO_2_):$$\;\left(C_{3}S\;+{\;C}_{2}S\right)+{\;H}_{2}O\;\to \;C\;-\;S\;-\;H\;+\;\mathrm{CH}\;+{\;\mathrm{SiO}}_{2}\;\to \;C\;-\;S\;-\;H$$

The product of the C-S-H pozzolanic reaction not only improves the adhesion between the cement stone and the surface of fine aggregate grains, but also results in compaction of the concrete structure. Thus, the pozzolanic reaction has the double effect: the increase in the compressive strength of concrete and the decrease in the total pore volume in its structure. In addition, micro silica particles < 0.1 µm can fill voids created by free water in the matrix.

The rapid increase in compressive strength of concretes containing river sand was observed at the age from 3 to 28 days. At the same time, the increase of strength at 28-day age as compared to 3-day strength with the micro silica content of 7.5, 10, 12.5 and 15% from the mass amounted to 56.3%, 43%, 56.9%, and 56%, respectively. After 28 days, these concretes showed a slight increase in strength, which by 120 days reached values of 106, 108, 109, and 117 MPa, respectively. The highest compressive strength (137 MPa at the age of 120 days) was obtained with a micro silica content of 7.5% from the mass of the multicomponent binder. However, the decrease of the concrete compressive strength was observed with the larger increase of the micro silica content. The obtained results can be due to the fact that, with adding the larger amount of micro silica in place of sulfate-resistant cement, the part of it is formed without entering into a pozzolanic reaction with the formation of low-basic calcium hydro silicates strengthening the cement stone of concrete. In addition, their amount is reduced due to a decrease in the cement content in a multicomponent binder. Thus, the obtained experimental results showed that the optimal content of micro silica in the concrete mixture is 7.5% from the mass of the multicomponent binder, other things being equal.

### 3.3. Axial and Flexural Tensile Strengths at 28-Day Age

The axial tensile strength of the developed concretes using white and river sand as a fine aggregate was determined after 28 days of hardening of the samples (Figure 7). The results show that the above strength increases with the increase of the micro silica content in the multicomponent binder from 5 to 12.5% from the mass, and then decreases with its further growth to 15% from the mass. It is worth noting that white sand formulations show higher axial tensile strength than river sand formulations. The highest strength value of 11.9 MPa in axial tension was shown by the concrete containing white sand with 7.5% from the mass SF and 20% from the mass FA, which is by 78% higher than that of the test concrete sample containing river sand.

The tensile strength in bending of 28-day age concrete samples was 9.5–29.2 MPa (Figure 7). When the multicomponent binder part of sulfate-resistant Portland cement was replaced simultaneously by 7.5% from the mass SF with fly ash up to 40% from the mass the tensile strength in bending of concretes with river sand ranged from 9.5 to 12.3 MPa, which is lower as compared to other compositions of the developed concretes. The test results of concrete samples of nine compositions with white and river sand containing 20% from the mass in the composition of the multicomponent binder, fly ash, and micro silica in the amounts of 5, 7.5, 10, 12.5, and 15% from the mass, showed that, other things being equal, using white sand enables to obtain the concrete of tensile bending strength varying from 21.4 to 29.2 MPa, up to 3 times higher than that of the concretes using river sand.

## 4. Conclusions

Based on the results of the conducted research, the following conclusions can be drawn:With the use of the local Vietnamese raw materials, high-strength fiber-reinforced concretes can be obtained, of the high potential for use in the different constructions. The conducted studies have discovered that the highest compressive strength, equal to 137 MPa at 120-day age, as well as the axial tension and tensile bending equal to 11.9 MPa and 29.2 MPa, respectively, at 28-day age, were shown by the concrete composition with natural white quartz sand, contained in the multicomponent binder composition based on sulfate-resistant Portland cement 7.5% from the mass microsilica combined with 20% from the mass fly ash from thermal power plants. Thus, it was determined that the optimal content of micro silica and low-calcium fly ash in a multicomponent binder is 7.5% and 20% from the mass, respectively.It has been established that using the natural white sand with grain size from 5 to 1800 μm increases the concrete strength characteristics as compared to river sand. Therefore, it is efficient to use white sand for producing concretes of high strength characteristics. The above use will also contribute to protecting the local river sand resources from depletion, which is relevant for Vietnam.The use of local waste from the power and metallurgy industries in the form of fly ash from thermal power plants and micro silica as a partial replacement for Portland cement in multicomponent binder compositions reduces the carbon footprint of the cementing components production and helps to protect the environment from industrial waste pollution. In addition, it is beneficial in terms of saving power resources and reducing the concrete cost.

## Figures and Tables

**Figure 1 materials-15-08851-f001:**
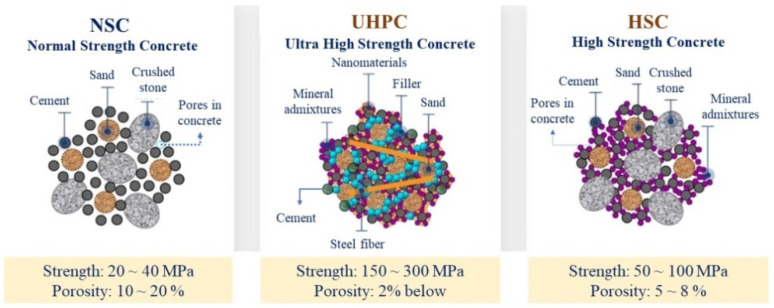
Concrete materials composition corresponding to various levels of characteristics for compression at the age of 28 days.

**Figure 2 materials-15-08851-f002:**
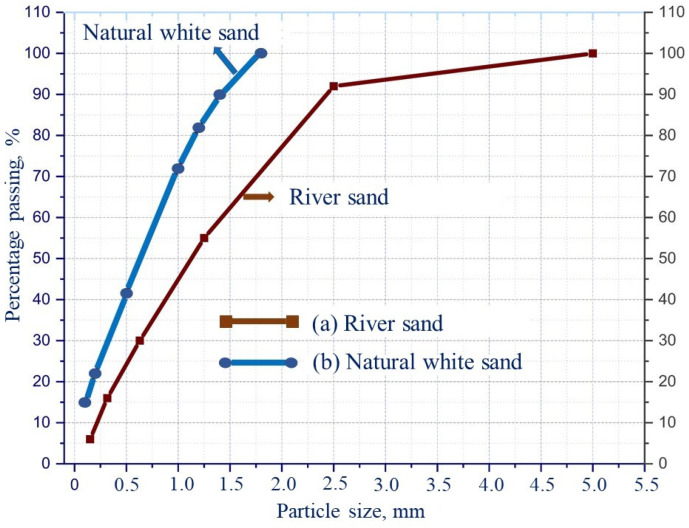
Particle size distribution curve for fine aggregate: (a) river sand; (b) natural white sand.

**Figure 3 materials-15-08851-f003:**
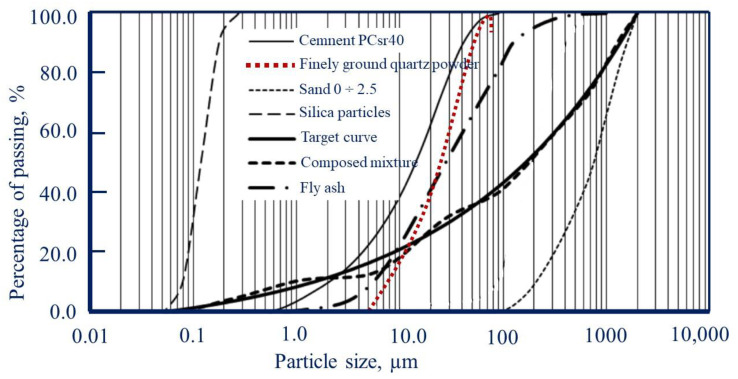
The particle size distribution of raw materials and fillers.

**Figure 4 materials-15-08851-f004:**
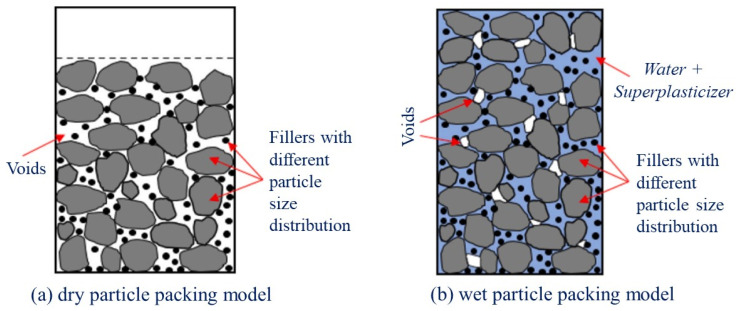
Particle packing models.

**Figure 5 materials-15-08851-f005:**
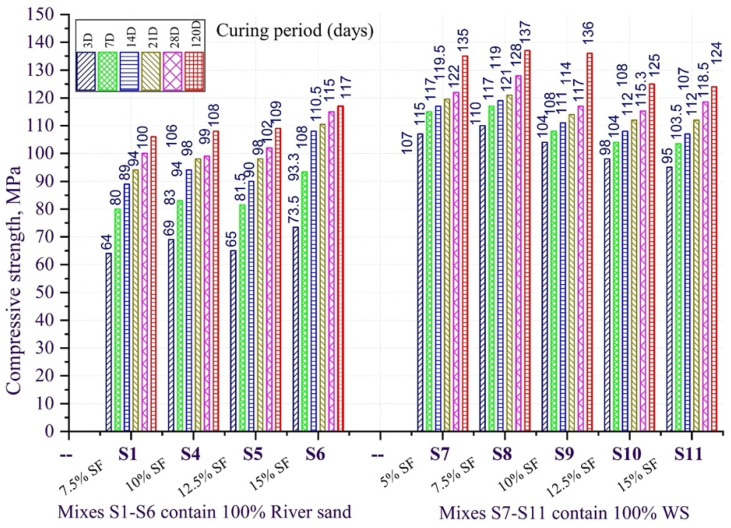
Compressive strength of concrete containing silica fume and fine aggregate with various content.

**Figure 6 materials-15-08851-f006:**
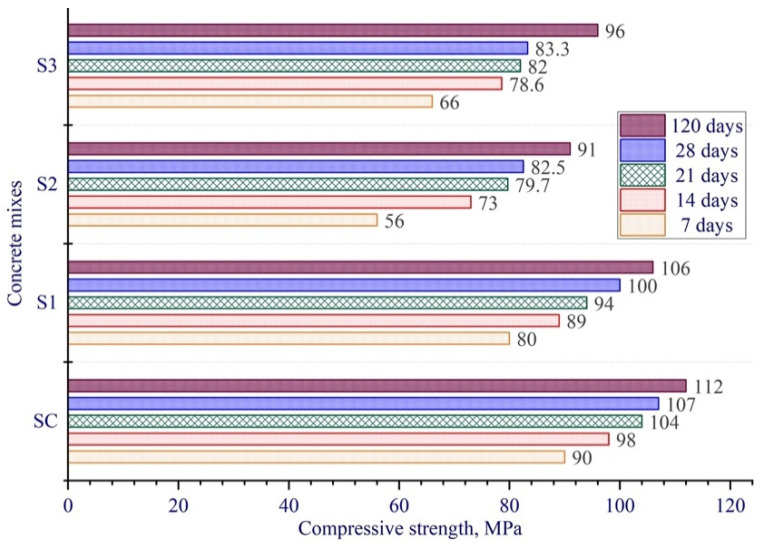
Compressive strength change of fly ash concrete.

**Figure 7 materials-15-08851-f007:**
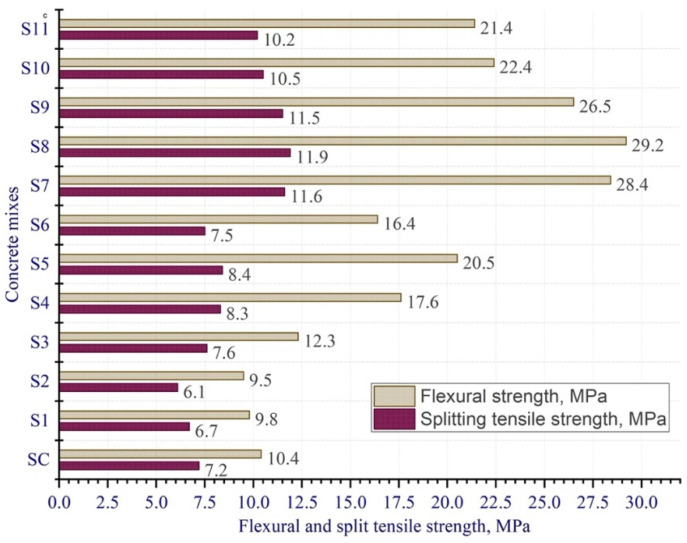
Splitting tensile and flexural strength of concrete at the age of 28 days.

**Table 3 materials-15-08851-t003:** Mix proportions of the developed UHPC mixtures (kg/m^3^ of concrete).

Concrete Mix №	Cementitious Materials, kg·m^−3^					Steel Fiber	Fine Aggregates, kg·m^−3^			Water	Superplast Icizer		Water/Binder
	Cement PCsr40	Silica Fume		Fly Ash			River Sand	White Sand	Ground Quartz				
	kg	%	kg	%	kg	kg	kg	kg	kg	Liter	%	Liter	-
SC	905	0	0	0	0	100	962	-	271	226	1.8	16.3	0.25
S1	656	7.5	68	20	181	100	962	-	271	226	1.8	16.3	0.25
S2	566	7.5	68	30	272	100	962	-	271	226	1.8	16.3	0.25
S3	475	7.5	68	40	362	100	962	-	271	226	1.8	16.3	0.25
S4	634	10	91	20	181	100	962	-	271	226	1.8	16.3	0.25
S5	611	12.5	113	20	181	100	962	-	271	226	1.8	16.3	0.25
S6	588	15	136	20	181	100	962	-	271	226	1.8	16.3	0.25
S7	679	5.0	45	20	181	100	-	962	271	226	1.8	16.3	0.25
S8	656	7.5	68	20	181	100	-	962	271	226	1.8	16.3	0.25
S9	634	10	91	20	181	100	-	962	271	226	1.8	16.3	0.25
S10	611	12.5	113	20	181	100	-	962	271	226	1.8	16.3	0.25
S11	588	15	136	20	181	100	-	962	271	226	1.8	16.3	0.25

## Data Availability

Not applicable.

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
