# Peer review of "The Effect of Mineral Admixtures and Fine Aggregates on the Characteristics of High-Strength Fiber-Reinforced Concrete"

_materials, 2022, doi:10.3390/ma15248851_

Round 1

Reviewer 1 Report

This work focuses on the mechanical properties of high-strength concrete. The topic is valuable and trendy. However, the organization and content of this manuscript is not satisfactory enough so that a further revision is required before its acceptance.

1. Some units are not appropriately presented, such as the cubic meters in Line 18. Please check the whole manuscript and make revisions.

2. The significance and contribution of this work is not stated clearly. The authors should clarify the current dilemma and situation in developing high-strength concrete and what are the contributions of this work to deal with the problems. In the abstract, there are some descriptions about the significance, which should be highlighted in the introduction part.

3. In the introduction, the authors mentioned silica fume, fly ash and fiber, however, they just reviewed the studies of silica fume (or micro silica). Considering the mineral materials and fibers are used together to produce high-strength concrete, the short review on other mineral materials such as fly ash and fibers is necessary. The following studies can improve, such as Influences of MgO and PVA fiber on the abrasion and cracking resistance, pore structure and fractal features of hydraulic concrete; Comparison between the influence of finely ground phosphorous slag and fly ash on frost resistance, pore structures and fractal features of hydraulic concrete; The influence of fiber type and length on the cracking resistance, durability and pore structure of face slab concrete.

4 The application of high-performance concrete should be placed at the very beginning of the introduction. This part, together with the first paragraph, should be an essential part to introduce the significance of your work and you should integrate these two parts together.

5 In figure 1, what kind of strength are involved? And what are the age when consider the strength?

6 In figure 4(a), the legends and coordinate are not presented in English. Please also check other figures to avoid such mistakes. Also, the figure 4(a) and (b) can be merged as one figure for better comparison.

7 In Figure 5, the cement is spelled wrongly.

8 The equations in the manuscript are in a messy way. Please make them uniform.

9 What does the lines mean in figure 10? Are these lines fitting curves or something else? It is not recommended to connect the discrete point without any assumptions or fitting process.

10 The figure 11 should be redrawn. The choice of abscissa is not appropriate. The authors should present the results with fine and river aggregate, respectively, while at the same time take the content of silica fume into considerations.

11 The abstract part should not be divided into several parts, one paragraph is suitable for the abstract.

Author Response

  1. Answer: the remark was taken into account, the required corrections were made to the text of the article.
  2. Answer: a description of the significance and contribution of the work to solving the problem of designing high-strength concrete compositions has been added to the introduction of the article, as well as a brief overview of current world trends in solving this problem
  3. Answer: we do not agree with the reviewer's statement that the article only considers the use of microsilica to obtain high-strength concrete. The introduction of the article also discusses the use of fly ash from thermal power plants, quartz flour and superplasticizers for this purpose.
  4. Answer: the reviewer's remark was taken into account, corrections were made to the text of the article.
  5. Answer: Obviously, this is not Figure 1, but Figure 2 in the new edition of the article. This figure refers to the compressive strength of concrete at a hardening age of 28 days, which has been clarified in the figure caption.
  6. Answer: The reviewer's remark about the need for the axes in English in Figure 3(a) has been taken into account in the new edition of the article and the necessary corrections have been made to this figure.
  7. Answer: note corrected.
  8. Answer: the reviewer's comment was taken into account, and the quality of writing the equations was improved.
  9. Answer: the reviewer's remark was taken into account and the graphic dependence shown in Figure 9 in the new edition of the article was replaced with a bar chart.
  10. Answer: the reviewer's remark was taken into account and corrections were made to Figure 10 in the new edition of the article.
  11. Answer: the reviewer's remark was taken into account, and the structure of the abstract part of the article was changed in accordance with the reviewer's recommendation.

Reviewer 2 Report

Paper ID: materials-2049340

Type: Article 
Title: The effect of mineral admixtures and fine aggregates on the characteristics of high-strength fiber-reinforced concrete

Authors: Olga Vladimirovna Aleksandrova , Quang Vinh Nguyen Duc , Boris Igorevich Bulgakov , Sergey Viktorovich Fedosov , Nadezhda Alekseevna Lukyanova , Victoria Borisovna Petropavlovskaya

  This study investigates the effect of the complex of active mineral additives consisting of silica and fly ash, and a fine aggregate, including finely ground natural - white quartz sand (BP) for partial replacement of river sand (RP), on the mechanical properties of high-strength concrete containing steel fiber. Although the testing methods and compared results attained in the present study show the importance of the paper, The authors should address the following comments: 

  1. NOVELTY IN COMPARISON TO RECENT LITERATURE? NEED TO BE EMPHASIZED.
  2. The results in the paper might be more discussed by the relevant literature.
  3. Abstract: Please combine the paragraphs.
  4. Please revise all subscripts and superscripts.
  5. Line 143: Figure 1 will be Figure 3.
  6. Figure 4: Please remove outer box of Fig. Please use English in Fig. Please use “.” As separator.
  7. Equations: please use Math type or etc. Resolutions are very bad.
  8. Figure: Please use error bars for strength.
  9. There should be a space between number and unit. Please correct these errors in the paper.
  10. Throughout the text, there are some typos that must be eliminated.
  11. The conclusion part seems to be more like an experimental report rather than a scientific paper. I strongly suggest for authors present their conclusions more concisely, avoiding repetition of the obvious and simple results.

Author Response

  1. Answer: The novelty of the way proposed in the article to solve the problem of obtaining high-strength concrete using raw materials local to Vietnam is to replace scarce river sand, whose natural resources are inevitably declining, with natural white quartz sand, which is still little used by the local building materials industry. This is stated in the introductory part of the article.
  2. Answer: the authors of the article believe that the introduction of the article provides a fairly impressive overview of 40 scientific sources on the subject of the article, and this remark of the reviewer will be taken into account in future works.
  3. Answers: the remark was taken into account, and the correction was made.
  4. Answers: the remark was taken into account, and the correction was made.
  5. Answers: the remark was taken into account, and the correction was made
  6. Answers: the remark was taken into account, and the correction was made
  7. Answers: the remark was taken into account, and the correction was made
  8. Answers: the remark was taken into account, and the correction was made
  9. Answers: the remark was taken into account, and the correction was made
  10. Answers: the remark was taken into account, and the correction was made
  11. Answers: the comment was taken into account, and the changes recommended by the reviewer were made to the final part of the article.

Round 2

Reviewer 1 Report

The manuscript can now be accepted.

Reviewer 2 Report

The authors have made necessary changes, therefore the manuscript can be accepted as it is.